# Comparative Study of Starch Phosphorylase Genes and Encoded Proteins in Various Monocots and Dicots with Emphasis on Maize

**DOI:** 10.3390/ijms23094518

**Published:** 2022-04-20

**Authors:** Guowu Yu, Noman Shoaib, Ying Xie, Lun Liu, Nishbah Mughal, Yangping Li, Huanhuan Huang, Na Zhang, Junjie Zhang, Yinghong Liu, Yufeng Hu, Hanmei Liu, Yubi Huang

**Affiliations:** 1National Demonstration Center for Experimental Crop Science Education, College of Agronomy, Sichuan Agricultural University, Chengdu 611130, China; 13863@sicau.edu.cn (G.Y.); nomanshoaib@stu.sicau.edu.cn (N.S.); xieying89@sicau.edu.cn (Y.X.); liulun@stu.sicau.edu.cn (L.L.); nishbahmughal97@gmail.com (N.M.); huanghuanhuan@sicau.edu.cn (H.H.); 2State Key Laboratory of Crop Gene Exploration and Utilization in Southwest China, Sichuan Agricultural University, Chengdu 611130, China; ypli@sicau.edu.cn (Y.L.); 13958@sicau.edu.cn (Y.H.); 3College of Science, Sichuan Agricultural University, Chengdu 611130, China; 72008@sicau.edu.cn; 4College of Life Science, Sichuan Agricultural University, Ya’an 625014, China; junjiezhang@sicau.edu.cn; 5Maize Research Institute, Sichuan Agricultural University, Chengdu 611130, China; 13964@sicau.edu.cn

**Keywords:** comparative study, cytosolic starch phosphorylase, plastidial starch phosphorylase, starch phosphorylase

## Abstract

Starch phosphorylase (PHO) is a multimeric enzyme with two distinct isoforms: plastidial starch phosphorylase (PHO1) and cytosolic starch phosphorylase (PHO2). PHO1 specifically resides in the plastid, while PHO2 is found in the cytosol. Both play a critical role in the synthesis and degradation of starch. This study aimed to report the detailed structure, function, and evolution of genes encoding PHO1 and PHO2 and their protein ligand-binding sites in eight monocots and four dicots. “True” orthologs of PHO1 and PHO2 of *Oryza sativa* were identified, and the structure of the enzyme at the protein level was studied. The genes controlling PHO2 were found to be more conserved than those controlling PHO1; the variations were mainly due to the variable sequence and length of introns. Cis-regulatory elements in the promoter region of both genes were identified, and the expression pattern was analyzed. The real-time quantitative polymerase chain reaction indicated that PHO2 was expressed in all tissues with a uniform pattern of transcripts, and the expression pattern of PHO1 indicates that it probably contributes to the starch biosynthesis during seed development in *Zea mays*. Under abscisic acid (ABA) treatment, PHO1 was found to be downregulated in *Arabidopsis* and *Hordeum vulgare.* However, we found that ABA could up-regulate the expression of both PHO1 and PHO2 within 12 h in *Zea mays*. In all monocots and dicots, the 3D structures were highly similar, and the ligand-binding sites were common yet fluctuating in the position of aa residues.

## 1. Introduction

Maize (*Zea mays*) is one of the three major cereal crops globally and the largest in food crops, with a yield second to rice (*Oryza sativa*). The major part of the yield is known as starch, which is the most important form of reserve polysaccharide synthesized in cellular organelles in plants called plastids [1]. Starch biosynthesis involves a set of enzymes including ADP-glucose pyrophosphorylase (AGPase; EC 2.7.7.27), starch synthases (SSs; EC 2.4.1.21), branching enzymes (SBEs; EC 2.4.1.18), starch debranching enzymes [DBEs, 3.2.1.41; mainly isoamylases (ISAs)] and starch phosphorylase (PHO; EC 2.4.1.1) in the cereal endosperm [2,3]. In recent times, PHO has been investigated comprehensively due to its involvement in starch biosynthesis [4,5,6,7,8,9].

The PHO is found to be involved in the transfer of glucosyl unit from glucose-1-phosphate (Glc-1-P) to growing α-1-4 linked glucan chain in a reversible reaction that depends on the available substrate level [10]. In plants, two distinct forms of PHO have been reported [11]: plastidial starch phosphorylase (PHO1) and cytosolic starch phosphorylase (PHO2). These two forms have significant sequence similarities to each other but largely differ in their molecular sizes, substrate specificities physiological roles, and intracellular localization [12]. PHO1 resides in the plastid [13] and PHO2 in the cytosol. PHO1 plays an important role only in starch degradation due to the high Pi/Glc-1-P ratio found in vivo [14]. However, no direct evidence is available to support this. Moreover, several starch-degrading enzymes have been identified [15,16], including glucan water dikinase (GWD; EC 2.7.9.4) and phosphoglucan water dikinase (PWD; EC 2.7.9.5), which produces the branched and linear glucans from the starch [17]. However, α-amylase (EC 3.2.1.1) is thought to be the first enzyme to initiate starch degradation. These findings make the PHO1 an indirect-acting or a regulatory candidate that can influence the activity of other participating enzymes. The exact role of PHO1 is still in debatable. Moreover, PHO1 is reported to be involved in starch metabolism in various monocots and dicots, including *Zea mays* [18], *Oryza sativa* [19] *Triticum aestivum* [4], *Hordeum vulgare* [20], and *Solanum tuberosum* [21]. PHO2 can degrade the branched glucans and can even attack the starch granules [22]. In addition, it can use the degraded products from the starch and maintain the Glc-1-P level in the cytosol [23].

These two enzymes are highly similar to each other, however they differ in the presence of transit peptides in the N-terminal and an extra stretch of 78–82 amino acid (aa) residues (L80 domain) in the middle region of PHO1. The presence of this extra peptide can be responsible for varied biochemical activities [9]. Previously, L80 insertion is found to be an active substrate for proteasomes in *Ipomoea batatas* [8]. The PHO1 in *Solanum tuberosum* maintains its intact structure in young tubers, however steadily degrades into smaller peptides as the tubers become mature [13]. Both the intact (105 kDa) and degraded (55 kDa) forms of PHO1 have been reported in *Vigna radiata*. Interestingly, both forms were identified to make active catalytic complexes [24]. In contrast, the PHO1 of cereals, including *Oryza sativa* [19,25], *Zea mays* [18], *Hordeum vulgare* [26], and *Triticum aestivum* [10] is not degraded by the proteasome, even though they all have a PEST motif in L80 insertion. A detailed comparative study would be required to uncover these differences to reach a conclusion. No major differences exist between PHO1 and PHO2 in terms of binding Glc-1-P and Pi (inorganic phosphate). However, these can be distinguished by their affinity to bind the glucan substrate (PHO1 has a low and PHO2 has a high affinity for highly branched polysaccharides) [19,27]. The L80 domain in PHO1 was found to sterically prevent the binding of the enzyme to the branched polysaccharide substrate [28]. Quantitative data about PHO abundance and activity is lacking and as of yet only *Hordeum vulgare* PHO1 crystal structure has been characterized [26].

Several reports on microarray based analysis combined with promoter sequences indicated that *PHO* may play a role under different stresses. However, the regulation pattern found variables among the main cereal crops, especially in *Arabidopsis thaliana*, where *PHO1* seemed to play a role in the tolerance of abiotic stress rather than the degradation of starch [29]. A study on *PHO1* of *Hordeum vulgare* suggested no effect of ABA on the expression of *PHO1*, although there was up-regulation of *PHO2* [20]. In contrast, in *Arabidopsis thaliana* significant down-regulation for both isoforms were detected. Considerable information about PHO and its gene is available. However, detailed information about gene diversity, structure variation, function, evolution, and PHO responding to phytohormones is unclear in some important crops, including *Zea mays*. In recent years, significant progress has been made by scientists in sequencing plant genomes. The availability of genome data is essential to answer the biological questions through comparative analysis [30]. Therefore, the present study was carried out using the PHO1 and PHO2 genes of *Oryza sativa* as a reference (as these are well characterized and closely related to *Zea mays*) to identify and characterize its orthologs from eight important monocots and four important dicots. The obtained results are presented in this communication, including “true” orthologs of *PHO1* and *PHO2* genes, evolution of gene structure for both isoforms across mentioned species, synteny and collinearity analysis, identification of conserved and variable domains with subtle differences between both isoforms and among the monocots and dicots, promotor analysis, and expression analysis with the effect of ABA treatment on the expression of both isoforms of PHO in *Zea mays*.

## 2. Results

### 2.1. Gene Sequence and Structure

#### 2.1.1. Identification of Orthologs of PHO1 and PHO2

The “true” orthologs of the PHO1 and the PHO2-encoding genes of *O. sativa* were identified from the 12 different species using the criteria mentioned in the Materials and Methods section. Thus, the genome, coding DNA sequence (CDS), and cDNA of PHO1 and PHO2 from the mentioned orthologs were obtained and are presented in Table 1 (PHO1) and Table 2 (PHO2). In all examined species, a single ortholog for each PHO1 and PHO2 was found. The similarity of cDNA of PHO1 in all mentioned orthologs with that of *O. sativa* (used as reference) ranged from 78.1% to 81.7% in monocots and 42.2% to 65.0% in dicots. Likewise, the similarity of cDNA of PHO2 ranged from 80.4% to 81.7% in monocots and 62.9% to 65.9% in dicots. The CDS similarity of PHO1 ranged from 90.4% to 91.9% in monocots and 80.5% to 81.6% in dicots. Likewise, the CDS similarity of PHO2 ranged from 86.0% to 93.5% in monocots and 82.8% to 85.1% in dicots. The similarity index of gene sequences was low compared with that of cDNA and CDS; it ranged from 49.2% to 63.0% in monocots and 45.9% to 46.3% in dicots; for PHO2 it ranged from 36.7% to 62.4% in monocots and 11.8% to 47.6% in dicots.

#### 2.1.2. Gene Structure Comparison

The gene structures for PHO1 (*PHO1*) and PHO2 (*PHO2*) from different species were compared using *O. sativa* as a reference. The length of the genes was a major difference, which ranged from 2.92 Kb to 9.02 Kb in *PHO1* and 5.24 Kb to 34.54 Kb in *PHO2*. *PHO1* was the longest for *Z. mays* and smallest for *S. tuberosum* (Table 1). Likewise, *PHO2* was the longest for *S. lycopersicum* and smallest for *Arabidopsis. thaliana* (Table 2). The variation in gene size was primarily attributed to the variations in size and number of introns. While comparing cDNA, introns were found to be lacking in 1.828–5.499 Kb in *PHO1* and 2.072–12.201 Kb in *PHO2*. cDNAs for both *PHO1* and *PHO2* were longer in dicots than in monocots. In CDS comparison, only marginal differences were observed for both *PHO1* and *PHO2* in monocots and dicots except for the *PHO1* of *A. thaliana*, where the CDS length was the shortest (compared with CDS of *PHO1* in all species) and *PHO2* of *P. hallii*, where the CDS length was the longest (compared with CDS *PHO1* in all species). Therefore, the detected cDNA variations were due to the length of untranslated regions (UTRs) in the cDNAs.

The number of introns and exons also displayed some differences among *PHO1* and *PHO2* of the examined species (Figure 1). Each *PHO1* had 15 exons and 14 introns in both monocots and dicots (except for the *PHO1* of *S. tuberosum*, which had seven exons and six introns). However, each *PHO2* had 15 exons with 14 introns (except for *PHO2* of *S. bicolor*, which had 16 introns) in monocots and 16 exons with 15 introns in dicots.

The difference in the size of introns and exons was also observed. The introns in *PHO1* were generally shorter than those of introns in *PHO2* where these were found to be larger in size ranging from 77 to 4638 bp (except for the first intron of *PHO1* in monocots). The intron 11 of the *PHO1* of *Z. mays* was found to be the longest among all *PHO1* of monocots and dicots (2092 bp). Likewise, the intron 13 of the *PHO2* of *S. tuberosum* was found to be the longest among all *PHO2* of monocots and dicots (4638 bp). The variations in the exon length were found to be relatively low, ranging from 77 to 476 bp for *PHO1* and 77 to 492 bp for *PHO2*. The Exon 12 of the *PHO1* in *S. lycopersicum* was the shortest (77 bp), whereas the exon 15 was the longest (476 bp). Likewise, the exon 4 of *PHO2* was the shortest (77 bp) and the exon 15 of *PHO2* was the longest (492 bp) in *O. sativa*.

Each type of intron phase was found in the *PHO1* of all monocots and dicots. However, only intron phase 0 and intron phase 2 were found in the *PHO2* of all monocots and dicots (Figure 1). An identical intron phase pattern was observed in both *PHO1* and the *PHO2* (except the PHO1 of *S. bicolor*). For both *PHO1* and *PHO2* phases, phase 0 was the most dominant in all the examined species, ranging from 71.42–78.57% for *PHO1* and 78.57–80% for *PHO2*. The average sequence similarity for exons in *PHO1* was found to be higher in monocots (82.84–85.31%) than in dicots (33.18–69.50%). However, the average sequence similarity for introns was low in both monocots (36.15–39.40%) and dicots (23.23–33.47%) (Figure 2). For *PHO2* exons, the similarity ranged from 84.50% to 86.17% in monocots and 19.21% to 31.75% in dicots; whereas, for introns, it ranged from 41.1% to 50.1% in monocots (except for *S. italica* where this was 72.8%) and 20% to 32.1% in dicots (Appendix A).

#### 2.1.3. Synteny and Collinearity Analysis

The synteny conservation analysis was carried out using the chunk of 31 genes with 15 genes flanking on either side of the *PHO1* and *PHO2* of *O. sativa* on chromosome number 3 and 1, respectively. For *PHO2,* some degree of synteny conservation was observed for *Z. mays, S. italica, T. aestivum, H. vulgare, B. distachyon,* and *S. bicolor*. However, for *PHO1*, some degree of synteny conservation for only *Z. mays*. and *P. hallii* was observed. The synteny of 12 of the 30 genes flanking on the *PHO1* of *O. sativa* was shared on the parallel chromosome 1 in *Z. mays* and chromosome 9 in *P. hallii*. Similarly, the synteny of 16 of the 30 genes flanking on *O. sativa PHO2* was shared on the parallel chromosome 3 in *Z. mays*, 16 of the 30 genes on chromosome V in *S. italica*, 19 of the 30 genes on chromosome 3A in *T. aestivum*, 17 of the 30 genes on chromosome 3H in *H. vulgare*, 19 of the 30 genes on chromosome 2 in *B. distachyon*, and 21 of the 30 genes on chromosome 3 in *S. bicolor*. The microsynteny analysis in terms of the nucleotide for *PHO1* and *PHO2* has already been described earlier in the gene structure comparison section (Appendix A).

#### 2.1.4. SSRs and Retro-Elements in PHO1 and PHO2

Both the *PHO1* and the *PHO2* were found to have the SSRs; but with variations in numbers and sequences of bp (Appendix A). No retrotransposons (LINEs and LTR elements), SINEs and transposons were identified in any of *PHO1* and *PHO2* of monocots and dicots.

#### 2.1.5. Promoter Analysis of PHO1 and PHO2

Cis-regulatory elements were identified through promotor analysis of about 1 kb upstream region of both *PHO1* and *PHO2* in each of the monocots and dicots. These regulatory elements (e.g., ABRE and MBS) apparently responded to hormones such as abscisic acid (ABA) and abiotic stress. Regulatory elements that induced the tissue-specific expression such as “endosperm expression” were also identified (Figure 3). TATA box and CAAT elements were common in both *PHO1* and *PHO2*. The effect of ABA regulatory elements was further analyzed in both *PHO1* and *PHO2* of *Z. mays* by employing the ABA treatments to cDNA (elaborated in the expression analysis section).

### 2.2. Protein Sequence and Structure

#### 2.2.1. Comparison of Protein Sequence

An extra stretch of about 78–82 aa residues was found in the middle region of PHO1 and not in PHO2; while for both PHO1 and PHO2, the length of the protein sequences was almost similar in all monocots and dicots, ranging 962–985 aa in PHO1 and 832–897 aa in PHO2 (Table 1 and Table 2). The extra peptide in the middle region of PHO1 was designated as the “L80 domain”. The comparison of aa sequences of both the PHO1 and the PHO2 in all monocots and dicots concerning the PHO1 and PHO2 of *O*. *sativa* showed higher similarity for PHO2 (89.2–90.4% in monocots and 75.0–77.8% in dicots) than for PHO1 (84.3–87.5% in monocots and 67.0–69.3% in dicots). PHO2 is found to be more conserved than PHO1.

#### 2.2.2. Sequence Variation in PHO1 and PHO2

Among all monocots and dicots, PHO1 was longer in *S. bicolor* than in other species. PHO1 had two conserved domains (CDs): GT-35-Glycogen-Phosphorylase (97–950 aa) and a unique L80 domain (500–600 aa). The L80 domain was hyper-variable and predicted to consist of a PEST motif (a motif of negatively charged aa residues, which could act as a substrate for proteasomes) [31,32]. Similarly, the N-terminal (1–97 aa) was also hyper-variable, although the linker segments were conserved (Figure 4). Insertions, deletions, and mismatches were the main reasons for variations. Each of the PHO1 was found with the insertion of a variable transit peptide of up to 50 amino acids, which lacked in PHO2 except for the PHO2 of *P. halli* (Appendix A). The transit peptide contained several structural features common to transit peptides of chloroplast proteins, however, contained a surprisingly large number of histidine residues.

PHO2 was relatively longer in *P. hallii* than in any other species. Unlike the lower conservation for PHO1, aa residues in PHO2 were more conserved for each of the 12 examined species. However, just like PHO1, PHO2 also possessed hyper-variable N-terminal residues in all species (Figure 4). PHO2 was also associated with the GT-35-Glycogen-Phosphorylase domain (45–830 aa), however unlike PHO1, it lacked the L80 domain in the middle region. Sequence variations were also observed for PHO2.

#### 2.2.3. Three Dimensional (3D) Structure Comparison of PHO1 and PHO2

The three-dimensional (3D) structures of PHO1 and PHO2 for all 12 species were generated using the PHO1 of *H. vulgare* (PDB id: 5lr8.1.A) and the PHO2 of *A. thaliana* (PDB id: 4beq.1.A) as templates, respectively. All generated structures had a high level of confidence, as evident from the following observations (Appendix A): (a) relative to allowed and disallowed regions, a high proportion of aa residues were present in the favored regions of the Ramachandran plot; (b) the value of G-factor was within the standard range for all comparisons; (c) the quality factor values calculated using ERRAT and VERIFY3D were also high; and (d) the Qmean and Dfire energy values (Swiss-model assessment–based values) were negative, suggesting the favorable energy environment for given aa residues. Both PHO1 and PHO2 were in homo-dimeric forms and almost had similar structures, except the L80 domain arranged outside the main core (Figure 5).

Different values for various parameters obtained after superimposition are mentioned in Appendix A. A high similarity level was observed upon pairwise alignment of 3D structures of PHO1 and PHO2 from each of the 12 species with the reference structure of the *O. sativa* ranging from 85.23% to 94.34% for PHO1 and 89.12% to 94.34% for PHO2. In addition, a high similarity level of intrinsic physical and chemical properties was also observed, ranging from 56.56% to 97.94% for PHO1 and 70.45% to 99.27% for PHO2 in each of the mentioned species. For PHO1, the root mean square deviation (RMSD) value was 2.04–5.68 Å in monocots and 2.10–3.87 Å in dicots. For PHO2 this was 0.04–1.63 Å in monocots and 0.25–0.37 Å in dicots. The main protein core regions of PHO1 and PHO2 were favorably superimposed to the reference structure except for the L80 domains in PHO1 (Figure 5, Appendix A).

#### 2.2.4. Ligand Binding Sites Analysis

The aa residues constituting the ligand-binding sites were identified only in PHO1 except for the PHO1 of *C. annuum*, *S. tuberosum*, *B. distachyon*, and *S. lycopersicum*. However, no ligand binding sites were identified for PHO2 except in C. annum (Figure 5 and Appendix A). The ligand-binding sites were confined to the residues in the GT-35-glycogen-phosphorylase domain and were nearer to the C-terminal. Generally, one to two clusters of ligands were predicted for the binding of pyridoxal-5-phosphate (PLP) molecules. For PHO1, aa residues ranging from 15 to 16 and from 14 to 15 were involved in the binding of the ligands in monocots and in dicot, respectively (similar for PHO2). In all monocots and dicots, the ligand-binding sites were common (∼99%), however, they fluctuated in the position of aa residues.

### 2.3. Phylogenetic Analysis

Initially, three separate trees were constructed for phylogenetic analysis based on aa sequences using PHO1, PHO2, and PHO1 + PHO2 for each. All trees were mostly identical. Only the tree based on PHO1 + PHO2 was performed (Figure 6). The tree had two major clades (PHO1 and PHO2), each with two main clusters including eight monocots and four dicots for both PHO1 and PHO2. The PHO1 and the PHO2 of *Z. mays* shared closer relationships with *S. italica*, *S. bicolor*, and *P. hallii* among monocots and were grouped in a subcluster.

### 2.4. Expression Analysis of PHO1 and PHO2

The results based on the microarray analysis showed that the expression pattern of PHO1 and PHO2 remained similar among all monocots and dicots. However, the expression pattern in the PHO1 of *Z. mays* was much more complex. In *Z. mays*, real-time quantitative polymerase chain reaction (RT-qPCR) analysis indicated that the expression level of PHO1 was the highest in the endosperm and the lowest in the anthers. Unlike PHO1, the expression level of PHO2 was the highest in the embryo and the lowest in the pollens. The results indicated that the expression level of PHO1 was low in the early stages of seed development, highest at 12 days after pollination, and then decreased until the end of the seed development. However, the expression level of PHO2 remained the same throughout the seed development and peaked at 24 DAF (Figure 7).

#### Effect of ABA Treatment on the Level of Expression of PHO1 and PHO2

Responses of PHO1 and PHO2 under ABA treatment varied considerably among the main cereal crops. Based on microarray data under different stress conditions, PHO1 was found to be downregulated by ABA; however, ABA only significantly altered the expression of PHO2 in the main cereal crops. Unlike the PHO1 of *H. vulgare*, the PHO1 of *Z. mays* was significantly up-regulated under ABA treatment just as PHO2 (Figure 8).

## 3. Discussion

Among both isoforms, PHO1 has been intensively investigated due to its main role in the growth and development of plants. In the present study we isolated the *PHO1* and *PHO2* of *Z. mays* and characterized their gene and protein structures together with evolutionary analysis in several other monocots and dicots. We analyzed their expression pattern based on RT-qPCR and microarray data and demonstrated the effect of ABA on the expression of *PHO1* and *PHO2* transcripts. Our results demonstrated the conservation of both isoforms of PHO, not only at the protein level, but also at the gene level in all the monocots and dicots that were characterized in this study. At the protein level, the conservation was evident by estimating similarity, identity, sequence coverage, presence of GT-35-glycogen-phosphorylase domain, and the 3D structure analysis across all the examined monocots and dicots. At the gene level, the exons were found to be more conserved than the introns in both isoforms of PHO across all demonstrated species. The expression of *PHO1* was found throughout the endosperm development suggesting that it could contribute with starch biosynthetic genes (SSs, SBEs and DBEs) during the seed development in *Z. mays*, as the availability of *PHO1* correlates with these enzymes [33,34,35], while, *PHO2* was found to be primarily expressed during seed germination. Various motifs were identified in the promoter regions of starch phosphorylase genes. The expression level of both isoforms of PHO under ABA treatment could be upregulated in *Z. mays* as suggested by the experimental results. In contrast, the expression level under different stress conditions varied much among several species.

PHO2 is found to be more conserved than PHO1 at both protein and gene levels. The higher similarity in PHO2 might be due to its functional conservation, which was nearly reported as similar in various species. Moreover, the variations in PHO1 could be attributed to its varying functions [8,28,36]. The regulatory and catalytic domains were found to be conserved in both PHO1 and PHO2 of all monocots and dicots. The ligand-binding sites were found to be mostly common for PHO, suggesting that their catalytic properties were closely related. Previously, it was described that a large portion of both of the PHO proteins shared by various species was similar and could be related to the function of phosphorylase [37]. Our results indicated that the PHO1 of the monocots was different from the PHO1 of dicots and PHO2 of all monocots and dicots in terms of an extra peptide in the middle (L80 region) and a transit peptide in the N-terminus. The addition of the extra peptide in the middle of PHO1 made it a differentially acting enzyme and could be critical in functional depiction [28]. Moreover, these regions were highly variable across the PHO1 of all monocots. The exact fate of this region has not been characterized yet. A specialized PEST motif [32] and various phosphorylation sites [31,38] have been described in this region of PHO1, which makes it critical in terms of structure, function, and regulation. Previously, phosphorylation of the starch biosynthetic enzymes was reported as the key modification for the regulation and activity of various enzymes [39]. The 3D modeling suggested the outside arrangements of the L80 domain from the main protein core and was found to restrict the activity of PHO1 for smaller polysaccharide substrates and sterically hinder the large polysaccharides, such as starch or glycogen [26]. Other than direct actions on the surface of the substrate, the L80 domain could also be involved in the phosphorylation-dependent complex formation [1]. Site-directed point mutations on phosphorylation sites, cleavage, or even complete removal of this L80 domain could increase the affinity of PHO1 for the larger molecules. Other than the L80 region and transit peptide in PHO1, both isoforms were mostly identical indicating that they might evolve from a single gene and the L80 domain was an additional product of an evolutionary process. The superimposition of predicted dimeric structures of PHO1 over the PHO1 of *H. vulgare* (only known structure of dimeric PHO1) and PHO2 over the PHO2 of *A. thaliana* (only known structure of dimeric PHO1) showed the accuracy of the structures and confirmed the identity of the true orthologs identified in the study.

The sequences of genes for PHO1 and PHO2 of monocots and dicots varied and were found to be more conserved in monocots. The number of introns and exons of *PHO1* remained constant during evolution, except in *S. tuberosum* (seven exons and six introns). An additional exon and intron were found in the *PHO2* of dicots. After the divergence of monocots and dicots from a common ancestor [40], exon loss occurred in monocots [41], which was supported by the fact that intron was longer in monocots than in dicots. The divergence of monocots and dicots was also apparent in our evolutionary tree. This is reasonable as storage organs for starch domestication patterns could vary in monocots and dicots. Higher variations in dicots suggested higher conservation in monocots than in dicots.

The lack of shared gene content (synteny) and gene order (collinearity) in most of the monocots and dicots, except in *Z. mays*, *S. bicolor*, *P. hallii* and *S. italica*, suggested that the chromosomes carrying the genes for PHO1 had undergone a massive reshuffling during evolution. The retention of some degree of synteny and collinearity in *S. bicolor*, *S. italica*, and *P. hallii* was understandable as these were more related to *Z. mays* compared with other evaluated species [42]. However, the lack of synteny and collinearity among the PHO1 of *H*. *vulgare*, *T. aestivum*, and *B. distachyon* was surprising as extensive conservation was reported in grasses [43], as seen in PHO2 of various species. The available literature on synteny and microsynteny [44,45] also suggested that synteny at the whole-genome level was conserved among grasses, however, it was missing in the genomic regions [44,45,46,47].

Several *cis*-regulatory elements have been prospected previously in the promotor region [32,37]. We also identified the elements responsible for the response to abiotic stress, tissue-specific expression, and light. The light-responsive element might have an important role as the starch synthesis took place during the daytime. Thus, a tight regulation pathway was required to control the response to light. The response elements in the upstream region of *PHO1* and *PHO2* might be responsible for the high expression level during the endosperm development. The prospected response elements in the present study need to be validated using wet-lab to determine the functional significance.

Microarray-based data have indicated the uniform pattern of *PHO1* transcripts in various tissues and seed developmental stages among *H. vulgare*, *O. sativa*, and *A. thaliana*. The higher expression of *PHO1* in the developing endosperm during the anthesis and grain filling period, and the lower expression in vegetative tissues, suggested its role in starch metabolism in *Z. mays*. Similar results were reported in *T. aestivum* [4], *O. sativa* [48], and *H. vulgare* [37]. The expression analysis of the *PHO2* of *Z. mays* suggested that it could mainly involve the degradation and maturation of starch, as the expression pattern was found to be relatively higher in the lateral stages of endosperm development. These results are agreeable with the PHO2 of *H. vulgare* [37] and *T. aestivum* [4] and it was shown that the expression of PHO2 correlates with its activity and it may degrade the reserve starch in plant organs [4]. In addition, it has been suggested previously that it may also be involved in metabolizing the products of starch degradation from chloroplast in cells with intact plastid, and regulates the cytosolic Glc-1-P level. The expression mode of *PHO1* was similar in the reported cereal crops, indicating that *PHO1* might play the identical role/activity in the metabolic pathways in each of the examined species, as the availability of *PHO1* correlates with the expression levels of other starch biosynthetic enzymes in higher plants [35]. The expression pattern of both isoforms are found to be relatively low in leaves for each of the examined species. Early speculation was that phosphorolysis was the major route of degradation in chloroplast [49], however, it is now clear that many chloroplast and starch containing plastids possess amylases in addition to PHO and can degrade the starch hydrolytically as well as phosphorolytically [50]. The importance of *PHO1* and *PHO2* in leaves have not been properly characterized. However, a study on the knockout mutant of *A. thaliana* (lack of *PHO1*) have found no alteration in starch and sugar metabolism and suggested that hydrolysis, rather than phosphorylsis, may be the major route of degradation in the chloroplast [51]. In addition, they show that phosphorolysis is not necessary for a normal metabolism in most leaf cells.

All studied *PHO1* and *PHO2* are proposed to have the regions (in promotors) that have functions in ABA-induced expression, but a study on PHO1 of *H. vulgare* suggested no effect of ABA on the expression of PHO1 but up-regulation of PHO2 [20]. However, in *A. thaliana* significant down-regulation for both isoforms were detected. In contrast, it is found that ABA can up-regulate the expression of both the *PHO1* and *PHO2* in *Zea mays*. The effect of abiotic stresses on the expression of PHO has not been determined for various crops and could be worthy of investigation. For *PHO1* and *PHO2* in maize, ABA-induced expression changes might be the same according to phylogenetic analysis and aa residue alignments.

## 4. Materials and Methods

### 4.1. Identification of “True” Orthologs of PHO1 and PHO2 Encoding Genes

Full-length cDNA and protein sequences of PHO in *O. sativa* were used as a reference in tBLASTx to identify “true” orthologs of PHO1 and PHO2 encoding genes in various monocots: *Zea mays* (*Z. mays*), *Setaria italica* (*S. italica*), *Panicum hallii* (*P. hallii*), *Brachypodium distachyon* (*B. distachyon*), *Oryza sativa* (*O. sativa*), *Hordeum vulgare* (*H. vulgare*) and dicots; *Arabidopsis thaliana* (*A. thaliana*), *Solanum lycopersicum* (*S. lycopersicum*), *Solanum tuberosum* (*S. tuberosum*), and *Capsicum annuum* (*C. annuum*) according to the criteria described in [52]. Briefly, the identified orthologs were based on a high level of sequence identities (more than 80% for monocots and more than 70% for dicots) and query the coverage along the PHO1 and PHO2 protein length, the presence of same domains and motifs as found in the original query sequence, and the conservation of sequence intervals between domains and motifs. Retrieved gene sequences were used to identify the full-length gene sequences from various databases, including NCBI (https://www.ncbi.nlm.nih.gov/, accessed on 5 January 2022), EMBL (https://www.embl.org//, accessed on 5 January 2022), and Ensembl (http://plants.ensembl.org/index.html/, accessed on 5 January 2022).

### 4.2. Gene Structure Analysis

Genomic DNA and CDS were used to determine the intron–exon junctions in full- length gene sequences. The intron phases (0, 1 and 2) were marked according to their position relative to the reading frame: intron insertion between two codons (phase 0), intron insertion after the first base pair of the codon (phase 2), or intron insertion after the second base pair of the codon (phase 3). Both exon and intron regions were employed to measure the GC content by GC calculator (https://www.sciencebuddies.org/, accessed on 1 January 2022). To evaluate the simple sequence repeats (SSRs) and retro-elements, a repeat masker 4.0.5 (http://www.repeatmasker.org/,accessed on 1 January 2022) was used with the default setting. The cis-regulatory response elements were identified in 1 kb upstream genomic regions of the translation start site using the PlantCare bioinformatics tool [53]. Following the criteria described by [54], the response elements on the sense and nonsense strand showing the matrix value of >5 was accepted.

### 4.3. Protein Sequence Analysis

All orthologs for PHO1 and PHO2 were used in multiple sequence alignment to generate the consensus amino acid (aa) sequence through DNAMAN software with default settings (http://www.dnaman.com, accessed on 1 January 2022). An aa present in all orthologs at a specific position was used in consensus sequence generation. Maize aa was used in consensus sequence if a different aa was found at the position. Insertions were also added to the consensus sequence that was relative to reference species [55]. All the aa sequences of orthologs were compared with the consensus sequence to find out the sequence similarity. A scale of 0–5 was used for the similarity index for the monocots and dicots. As such, zero value indicated the lack of similarity, whereas the value of five in monocots and the value of five in dicots suggested the conservation of aa compared with aa in the consensus sequence. CDD analysis was carried out for depicting various domains and motifs in the consensus sequence (http://www.ncbi.nlm.nih.gov/Structure/cdd/wrpsb.cg, accessed on 1 January 2022).

### 4.4. 3D Structure Analysis of the PHO1 and the PHO2

The Swiss model of the automated mode was used to generate the 3D structures of PHO1 and PHO2 using their aa sequences. The generated 3D structures were subjected to geometric and energetic verifications, and for this purpose, the following servers were used; structure analysis and verification server (SEVES) (http://nihserver.mbi.ucla.edu/SAVES, accessed on 1 January 2022), PROCHECK to check the aa in favored regions relative to other regions [56], ERRAT to evaluate the digits of non-bonded interactions among different atoms [57], VERIFY3D to evaluate the 3D structure compatibility with its aa sequence [58] and a structure assessment tool in the Swiss-model server. To confirm the PHO1 and PHO2 3D structures, the FATCAT server was used by superimposing the 3D structures of PHO1 and PHO2 from each of the mentioned species on the 3D structures of the PHO1 of *H. vulgare* and the PHO2 of *A. thaliana*, respectively.

### 4.5. Ligand-Binding Site Analysis

Scrutiny for the ligand-binding sites was conducted for both PHO1 and PHO2 using the 3DligandSite (http://www.sbg.bio.ic.ac.uk/3dligandsite/advanced.cgi, accessed on 5 January 2022) online software [59]. Since no output was available when PHO2 from the mentioned species was used, no analysis was carried out for PHO2.

### 4.6. Phylogenetic, Synteny and Collinearity Analysis

MEGA7 [60] was employed to analyze the phylogenetic relationship using the aa sequences of PHO1 and PHO2. The unrooted phylogenetic tree was constructed by the neighbor-joining method of distance matrix [61] with default settings. Synteny and collinearity of PHO1 and PHO2 were studied with Genomicus [62] (https://www.genomicus.bio.ens.psl.eu/, accessed on 1 January 2022) using blocks of genes associated with the PHO1 and PHO2 of *O. sativa*.

### 4.7. Preparation of Plant Material

Mo17 inbred line of *Z. mays* was selected to identify the expression level and the effect of ABA treatment on the expression level of PHO1 and PHO2. The inbred line was grown under the normal field conditions at Chongzhou Research Base of Sichuan Agricultural University, China. The materials were collected at 2-day intervals and preserved at −80 °C for experiments.

### 4.8. Expression Analysis

In silico expression analysis of the genes for PHO1 and PHO2 of the main cereal crops (*O. sativa*, *H. vulgare*, and *A. thaliana*) was carried out using the “Genevestigator” microarray database. In the case of *Z. mays*, the expression analysis in different tissues was carried out using the expression level data determined by RT-qPCR. The extraction of RNA, cDNA synthesis, and expression analysis were carried out as described by Jian M. [37]. The *GAPDH* gene encoding glycerinaldehyde-phosphate dehydrogenase was used as internal reference. The following primers were used to measure the transcript level of *PHO1*: Fwd: CTAACAGGACAATATGCA; Rev: GCTTCATTGGCCTTGGCA. The following primers were used to measure the transcript level of *PHO2*: Fwd: ATGAGTGCGGCGGCGAA; Rev: CTTGGCAATGCGCTGCTTCA. 

### 4.9. ABA Treatment Analysis

The middle seed of corn cob 15 days after self-pollination of Mo17 was placed in a liquid MS medium containing 50 μmol/L ABA to slowly shake (50 rpm) for 24 h. At the same time, the seeds were cultured in the MS medium without ABA as the control. Some seeds were taken out at indicated time points (0, 1, 4, 12, and 24 h), quickly frozen with liquid nitrogen, and stored for an RT-qPCR experiment.

## Figures and Tables

**Figure 1 ijms-23-04518-f001:**
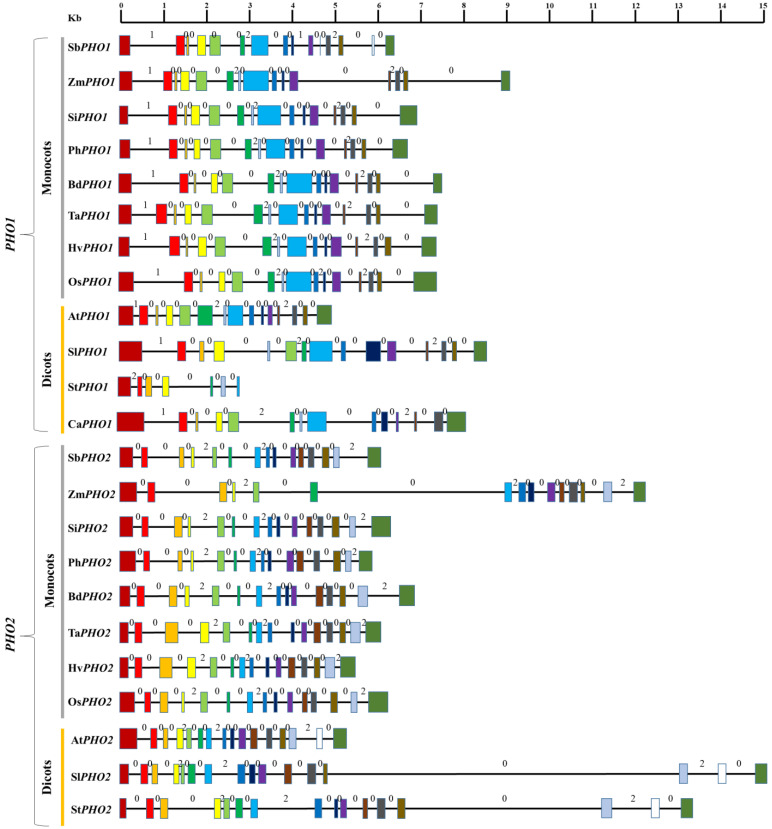
Gene structure of PHO1 and PHO2 from the translation start to stop sites in various monocots and dicots. Symbols include: Sb, *Sorghum bicolor*; Zm, *Zea mays*; Si, *Setaria italica*; Ph, *Panicum hallii*; Bd, *Brachypodium distachyon*; Ta, *Triticum aestivum*; Hv, *Hordeum vulgare*; Os, *Oryza sativa*; At, *Arabidopsis thaliana*; Sl, *Solanum lycopersicum*; St, *Solanum tuberosum*; and Ca, *Capsicum annuum*. Solid boxes indicate the exons, and lines indicate the introns. Colors represent the similarities among respective exons as a reference to *O. sativa*. The values of zero, one, and two marked above each intron are intron phases.

**Figure 2 ijms-23-04518-f002:**
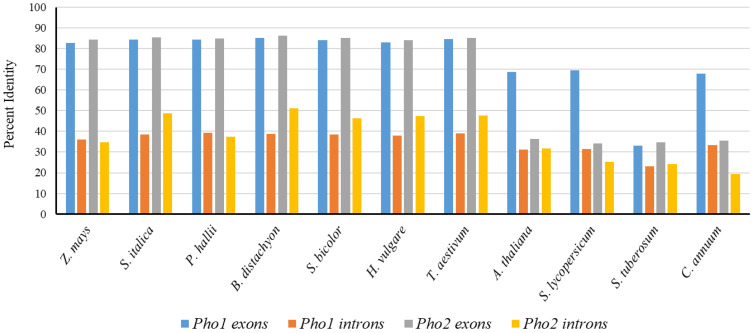
Percent identity of exons and introns in *PHO1* and *PHO2* of monocots and dicots with respect to exons and introns of *O. sativa*.

**Figure 3 ijms-23-04518-f003:**
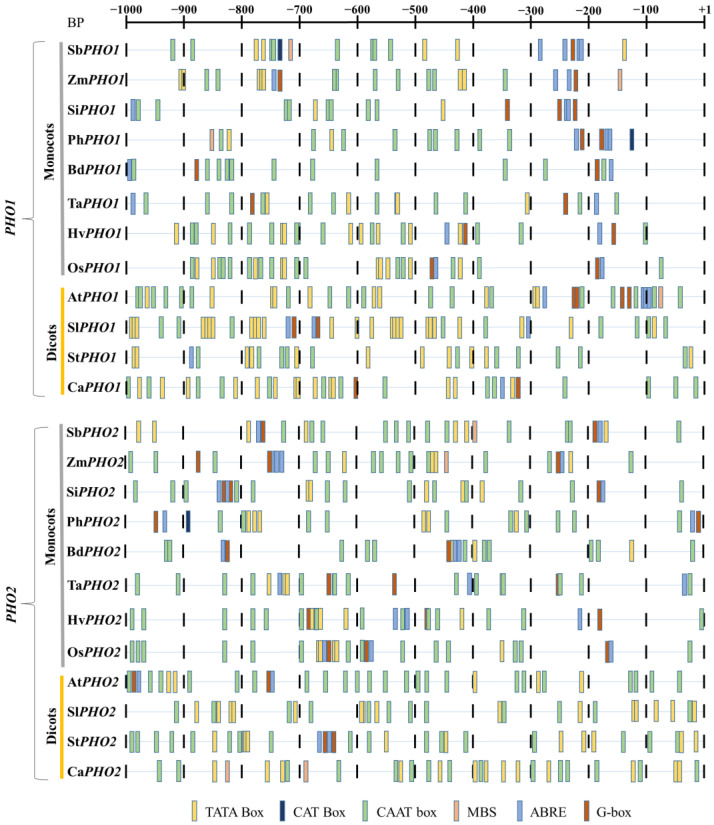
Representative figure showing regulatory elements identified in the 1 kb upstream region of *PHO1* and *PHO2*. Different color bars represent the major regulatory elements. Symbols include: Sb, *Sorghum bicolor*; Zm, *Zea mays*; Si, *Setaria italica*; Ph, *Panicum hallii*; Bd, *Brachypodium distachyon*; Ta, *Triticum aestivum*; Hv, *Hordeum vulgare*; Os, *Oryza sativa*; At, *Arabidopsis thaliana*; Sl, *Solanum lycopersicum*; St, *Solanum tuberosum*; and Ca, *Capsicum annuum*.

**Figure 4 ijms-23-04518-f004:**
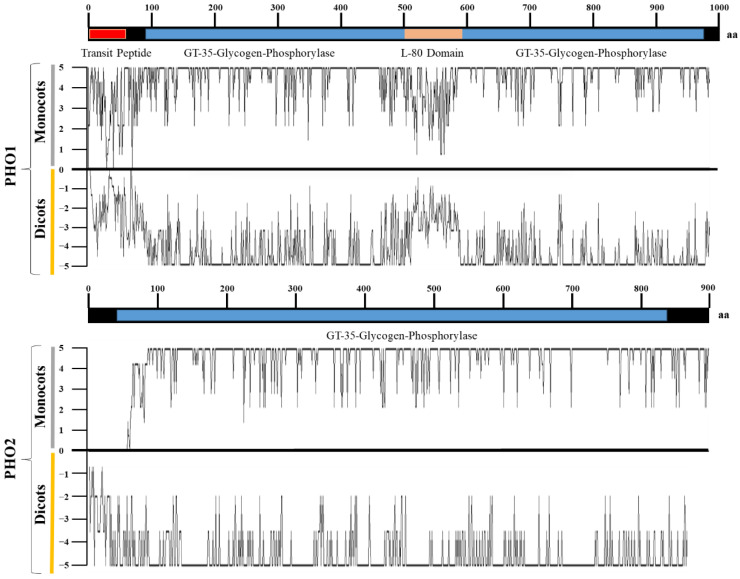
Amino acids sequence similarity of PHO1 and PHO2 among various monocots and dicots with respect to the consensus sequence. A value position of zero indicates the consensus sequence. Similar amino acids are plotted on scales 1–5 in monocots and −1 to −5 in dicots.

**Figure 5 ijms-23-04518-f005:**
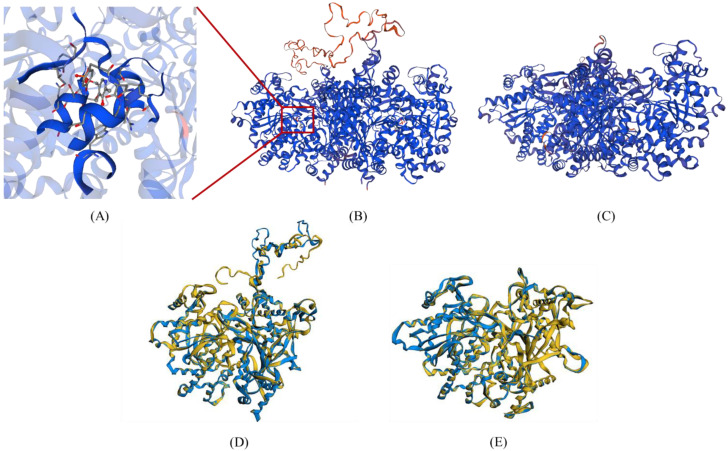
The 3D structures of maize PHO1 and PHO2: (**A**) Amino acid residues and their location involved in the binding of a ligand (PLP); (**B**) Homo-dimeric 3D structure of the PHO1 of *Z. mays*. The L80 domain (orange) arranged outside the main protein core; (**C**) Homo-dimeric 3D structure of the PHO2 of *Z. mays*; (**D**) Superimposed structures of the predicted PHO1 of *Z. mays* (yellow-colored) over the PHO1 of *H. vulgare* (blue colored); (**E**) Superimposed structures of the predicted PHO2 of *Z. mays* (yellow-colored) over the PHO2 of *A. thaliana* (blue colored).

**Figure 6 ijms-23-04518-f006:**
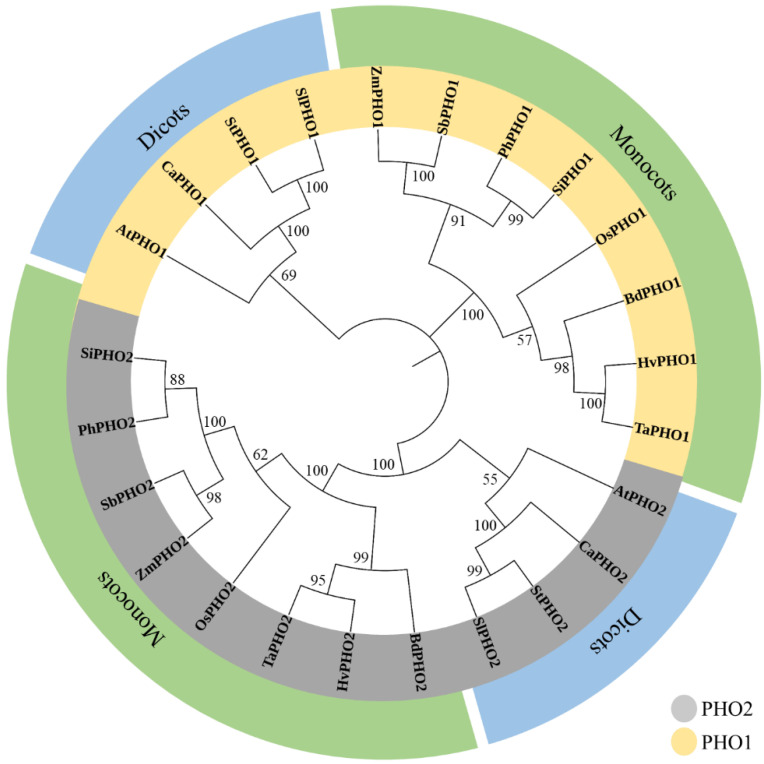
A phylogenetic tree constructed by the neighbor-joining method using the amino acid sequences of PHO1 and PHO2 to depict the relationship among monocots and dicots. The bootstrap value is calculated based on 1000 replications and displayed on each node. Symbols include: Sb, *Sorghum bicolor*; Zm, *Zea mays*; Si, *Setaria italica*; Ph, *Panicum hallii*; Bd, *Brachypodium distachyon*; Ta, *Triticum aestivum*; Hv, *Hordeum vulgare*; Os, *Oryza sativa*; At, *Arabidopsis thaliana*; Sl, *Solanum lycopersicum*; St, *Solanum tuberosum*; and Ca, *Capsicum annuum*. The branch length indicates the magnitude of genetic changes.

**Figure 7 ijms-23-04518-f007:**
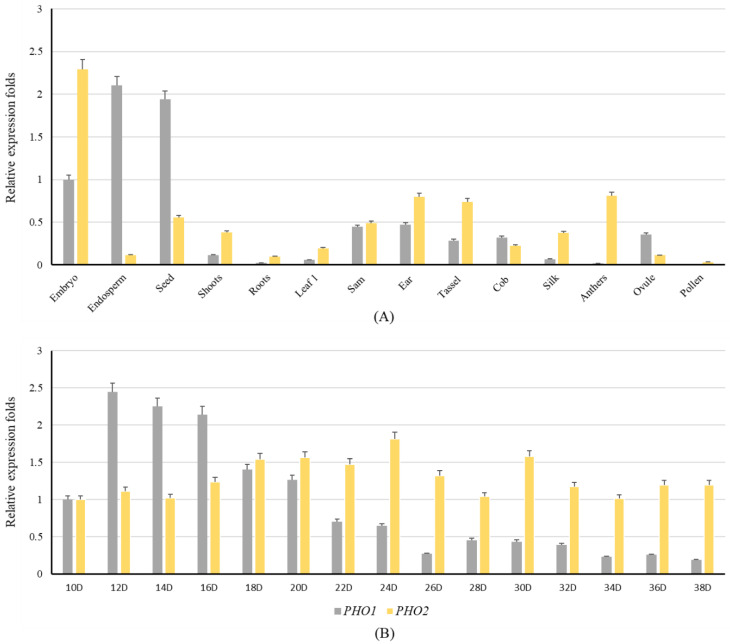
Distribution and expression pattern of the *PHO1* and *PHO2* transcripts of *Z. mays:* (**A**) Tissue-specific expression levels of the *PHO1* and *PHO2* of *Z. mays*. The relative expression pattern is shown and the transcript level in the embryo is used as control. The leaf1 is the first leaf harvested from the mature plant when it is flowering. The 15 DAP seeds and dissected endosperm were used to analyze the transcript level; (**B**) Relative expression levels of the *PHO1* and *PHO2* of *Z. mays* in different developmental stages of the endosperm. The transcript level in 10 DAP endosperm was used as a control.

**Figure 8 ijms-23-04518-f008:**
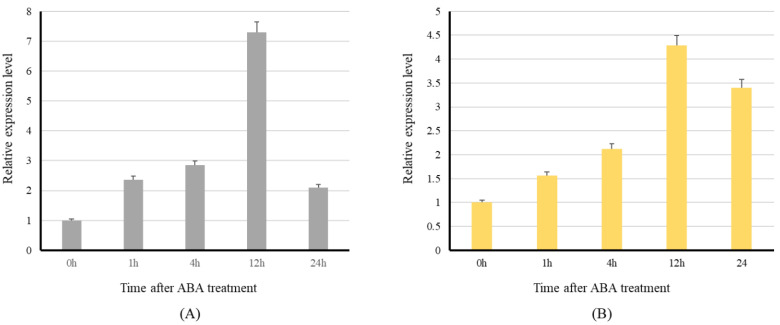
Expression level of the *PHO1* and *PHO2* of maize under ABA treatment. The 15 DAP middle seed of corn cob was used to treat with ABA in MS medium. Furthermore, 0 h indicates no ABA treatment and is used as a control. (**A**) Effect on *PHO1* transcripts expression. (**B**) Effect on *PHO2* transcripts expression.

**Table 1 ijms-23-04518-t001:** Details of cDNAs, CDS, genes and protein sequences of PHO1 in different monocots and dicots with respect to *O. sativa* PHO1.

Species PHO1	cDNA		CDS		Gene		* Protein	
	Length (bp)	% Identical	Length (bp)	% Identical	Length (bp)	% Identical	Length (aa)	% Identical
*S. bicolor*	3445	79.0	2958	91.9	6536	62.6	985	86.2
*Z. mays*	3523	78.1	2955	91.9	9022	49.2	984	85.8
*S. italica*	3385	80.9	2943	92.4	6909	62.8	980	87.2
*P. halli*	3351	80.7	29.64	91.9	6671	63.0	987	85.8
*B. distachyon*	3410	81.7	2937	92.5	7498	62.9	978	87.5
*T. aestivum*	3312	80.0	2917	90.4	7461	58.4	971	84.3
*O. sativa*	3408	100	2937	100	7239	100	978	100
*H. vulgare*	3305	77.3	2907	90.6	7441	61.1	968	84.9
*A. thaliana*	3180	64.5	2889	81.5	5008	45.9	962	69.3
*S. lycopersicum*	3332	65.0	2901	80.9	8718	46.3	966	67.0
*S. tuberosem*	4942	42.2	2901	81.6	4942	42.2	966	67.4
*C. annuum*	3456	63.2	2937	80.5	8233	45.9	978	68.5

* Accession ids for *S. bicolor*, XP_021306483.1; *Z. mays*, NP_001296783.1; *S. italica*, XP_004981704.1; *P. halli*, XP_025795811.1; *B. distachyon*, XP_003559211.1; *T. aestivum*, ACC59201.1; *O. sativa*, XP_015631420.1; *H. vulgare*, KAE8783983.1; *A. thaliana*, Q9LIB2.1; *S. lycopersicum*, NP_001362574.1; *S. tuberosem*, NP_001275215.1; *C. annuum*, XP_016569840.1.-indicates absence of data.

**Table 2 ijms-23-04518-t002:** Details of cDNAs, CDS, gene, and protein sequences of PHO2 in different monocots and dicots with respect to *O. sativa* PHO2.

Species PHO2	cDNA		CDS		Gene		* Protein	
	Length (bp)	% Identical	Length (bp)	% Identical	Length (bp)	% Identical	Length (aa)	% Identical
*S. bicolor*	3167	80.8	2517	92.5	6227	60.6	838	89.5
*Z. mays*	2868	80.4	2517	92.6	12,006	36.7	838	89.9
*S. italica*	2937	81.7	2514	92.4	6312	61.6	837	89.2
*P. halli*	2979	81.2	2664	86.0	5902	59.4	897	83.2
*B. distachyon*	2909	82.5	2502	93.4	6891	62.4	833	90.3
*T. aestivum*	2856	80.4	2499	93.2	6114	62.2	832	90.3
*O. sativa*	3036	100	2526	100	6275	100	841	100
*H. vulgare*	2857	79.4	2499	93.5	5569	59.6	832	90.4
*A. thaliana*	3168	62.9	2526	85.1	5240	47.6	841	77.6
*S. lycopersicum*	2908	65.4	2514	84.9	15,109	23.7	837	77.1
*S. tuberosem*	2833	65.9	2517	85.1	13,343	29.7	838	77.8
*C. annuum*	2965	65.0	2544	82.8	34,548	11.8	847	75.0

* Accession ids for *S. bicolor*, XP_002458773.1; *Z. mays*, NP_001151625.1; *S. italica*, XP_004970602.1; *P. halli*, XP_025814328.1; *B. distachyon*, XP_003564622.1; *T. aestivum*, Q9LKJ3.1; *O. sativa*, XP_015621275.1; *H. vulgare*, KAE8796869.1; *A. thaliana*, CAB61943.1; *S. lycopersicum*, XP_004246972.1; *S. tuberosem*, NP_001275118.1; *C. annuum*, XP_016542003.1.-indicates absence of data.

## Data Availability

Not applicable.

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
