# Peer review of "Comparative Study of Starch Phosphorylase Genes and Encoded Proteins in Various Monocots and Dicots with Emphasis on Maize"

_ijms, 2022, doi:10.3390/ijms23094518_

Round 1

Reviewer 1 Report

Review of manuscript by Yu et al (Comparative Study of Starch Phosphorylase Genes and En-coded Proteins in Various Monocots and Dicots with Emphasis on Maize) [ijms1636137]

The paper by Yu et al. is a fairly comprehensive analysis of plant starch phosphorylases (SP). The work details gene structure and protein structure as well as gene expression analysis of plastidial starch phosphorylase (PHO1) and cytosolic starch phosphorylase (PHO2) from a range of monocot and dicot plants. The paper represents a useful source of information the plant SPs. I have a few comments and suggestions for the authors to consider regarding the manuscript.

1. The abstract (and also introduction, line 55) states that PHO1 resides in the amyloplast. More accurately, PHO1 is a plastidial form of SP and is found in all plastids that make starch (e.g. chloroplasts and amyloplasts).

2. The abstract (and later, lines 310-312) mentions on line 29 that PHO1 expression is similar to that of other starch related genes in maize seed development. This needs rewording as the authors do not show this data or mention it directly. If expression is similar to other starch related genes from previous studies, this should be mentioned and cited.

3. Lines 34-37 are opinion and not relevant in an abstract. I suggest they are removed.

4. Reference 13 on line 61 of the introduction is a review and not primary research. Please replace with the original work citing this discovery.

5. Line 70 should read “...participating enzymes.”

6. Table 1 has a typographical error; it should read PHO1 (not PHO2)

7. Line 174, the term “.. promising..” is inappropriate, please re-write.

8. Line 202 should be re-written as it is confusing. It should read “....region of PHO1 and not in PHO2;....”

9. Details in Figure 1 legend should also be included in Figure 2 legend (rather than stating symbols are same as Fig. 1); otherwise the figure is hard for the reader to follow.

10. Line 217 states “.. which acted as...” this was not shown in the paper, and if it has been shown previously by other groups should be stated as such and cited properly. If this has not been shown, but is conjecture, then it should be stated clearly in the text as such.

11. The detailed analysis of amino acid sequences of both PHO1 and PHO2 enzymes (Figure 4) should include analysis and mention of the transit peptides associated with the plastidial PHO1 enzymes. Transit peptides, their structures/sequences and their positions in the various PHO1s is not mentioned and should be.

12. Line 248 “....high level of similarity for physico-chemical properties..”. This is confusing in terms of meaning and should be re-written more clearly.

13. Figure 7 outlines expression analysis of PHO1 and PHO2 in maize tissues. In Fig 7A more detail is required; please explain what leaf is leaf 1? Is it the first leaf which forms approx. 7days after germination or is it the first leaf harvested from the mature plant when it is flowering? What is the distinction between the ear, seed and endosperm? In Fig 7B the legend states that the data presented is for embryo. Is this correct, or do the authors mean endosperm. If embryo, why was this tissue analyzed and not the starch storing endosperm? If embryo was analyzed, there is no mention in the methods how it was dissected.

14. Figure 8 shows changes of PHO1 and PHO2 transcript by maize in response to ABA. What tissue in maize was used for these analyses and what developmental stage?

15. Line 343 is a typographical error, it should read “hinder”.

16. Please explain the statement on lines 386-388 as to why PHO2 is involved in starch degradation? There is no evidence presented in the current manuscript to support this statement. Lines 388-390 need to be explained. How can PHO1 and PHO2 play the same role in the (starch?) metabolic pathway as they are in different cellular compartments?

17. There is no explanation as to why both PHO genes are expressed at such low levels in leaves. Some mention of this is warranted, particularly as maize leaves produce starch which is turned over during a 24h cycle.

18. In the methods section, more detail is required in the section on gene expression analysis. What controls were used. This (control) data should be included.

Author Response

Response to Reviewer 1 Comments

Reviewer: The paper by Yu et al. is a fairly comprehensive analysis of plant starch phosphorylases (SP). The work details gene structure and protein structure as well as gene expression analysis of plastidial starch phosphorylase (PHO1) and cytosolic starch phosphorylase (PHO2) from a range of monocot and dicot plants. The paper represents a useful source of information the plant SPs. I have a few comments and suggestions for the authors to consider regarding the manuscript.

Dear reviewer,

Thank you for your careful reading and positive comments and suggestions on our manuscript. We have added a point-to-point response to your comments in the revised version of the manuscript. We felt that after the incorporation of changes you suggested, manuscript quality has been greatly improved. We have highlighted all the changes we made in red color.

  1. The abstract (and also introduction, line 55) states that PHO1 resides in the amyloplast. More accurately, PHO1 is a plastidial form of SP and is found in all plastids that make starch (e.g.chloroplasts and amyloplasts).

Yes, thank you for the correction we have made changes in the revised version.

Following changes have been made.

Abstract.

PHO1 specifically resides in the plastid, while PHO2 is found in the cytosol.

Line in revised version 54.

PHO1 resides in the plastid [12].

  1. The abstract (and later, lines 310-312) mentions on line 29 that PHO1 expression is similar to that of other starch related genes in maize seed development. This needs rewording as the authors do not show this data or mention it directly. If expression is similar to other starch related genes from previous studies, this should be mentioned and cited.

Sorry, it was a mistake. We have made the changes in the revised version of manuscript.

Following changes have been made.

Abstract

The real-time quantitative polymerase chain reaction indicated that PHO2 was expressed in all tissues with a uniform pattern of transcripts, and the expression pattern of PHO1 is indicating that it probably contributes to the starch biosynthesis during seed development in Zea mays.

Lines in revised version 333-337.

The expression of PHO1 was found throughout the endosperm development suggesting that it could contributes with starch biosynthetic genes (SSs, SBEs and DBEs) during the seed development in Z. mays, as the availability of PHO1 correlates with these enzymes [33-35]. While, PHO2 was found to be primarily expressed during seed germination.

  1. Lines 34-37 are opinion and not relevant in an abstract. I suggest they are removed.

Thank you for the suggestion, we have removed these lines.

  1. Reference 13 on line 61 of the introduction is a review and not primary research. Please replace with the original work citing this discovery.

Sorry for this wrong citation, the original citations are being added.

Following changes have been made.

However, these can be distinguished by their affinity to bind the glucan substrate (PHO1 has a low and PHO2 has a high affinity for highly branched polysaccharides) [19, 27].

  1. Line 70 should read “...participating enzymes.”

Sorry, it was a mistake.

Following changes have been made.

Line in revised version 62.

These findings make the PHO1 an indirect-acting or a regulatory candidate that can influence the activity of other participating enzymes.

  1. Table 1 has a typographical error; it should read PHO1 (not PHO2)

Sorry, it was a mistake the PHO2 is been replaced with PHO1 in Table 1.

  1. Line 174, the term “.. promising.” is inappropriate, please re-write.

Sorry for this mistake, following changes have been made in the revised version.

Line in revised version 193.

For PHO2, some degree of synteny conservation was observed for Z. mays, S. italica, T. aestivum, H. vulgare, B. distachyon, and S. bicolor.

  1. Line 202 should be re-written as it is confusing. It should read “....region of PHO1 and not in PHO2;....”

Thank you for the correction. Suggested changes have been mentioned in the revised version.

Following changes have been made (line in revised version

An extra stretch of about 78-82 aa residues was found in the middle region of PHO1 and not in PHO2.

  1. Details in Figure 1 legend should also be included in Figure 3 legend (rather than stating symbols are same as Fig. 1); otherwise the figure is hard for the reader to follow.

Thank you for the suggestion, we have included the symbols in figure legend.

Figure 3. Representative figure showing regulatory elements identified in 1 kb upstream region of PHO1 and PHO2. Different color bars represent the major regulatory elements. Symbols include: Sb, Sorghum bicolor; Zm, Zea mays; Si, Setaria italica; Ph, Panicum hallii; Bd, Brachypodium distachyon; Os, Oryza sativa; At, Arabidopsis thaliana; Sl, Solanum lycopersicum; St, Solanum tuberosum; and Ca, Capsicum annuum.

  1. Line 217 states “.. which acted as...” this was not shown in the paper, and if it has been shown previously by other groups should be stated as such and cited properly. If this has not been shown, but is conjecture, then it should be stated clearly in the text as such.

Sorry for this mistake, we have made changes in this line and citations have also been added.

Following changes have been made (lines in revised version 212-214).

The L80 domain was hyper-variable and predicted to be consists of a PEST motif (a motif of negatively charged aa residues, which could act as a substrate for proteasomes) [31, 32].

  1. The detailed analysis of amino acid sequences of both PHO1 and PHO2 enzymes (Figure 4) should include analysis and mention of the transit peptides associated with the plastidial PHO1 enzymes. Transit peptides, their structures/sequences and their positions in the various PHO1s is not mentioned and should be.

Thank you for the suggestion. We have added some detail in the passage in revised version. It was difficult to mention the sequential details in the figure 4 however; we have included the sequential features in the supplement figure (Figure S3).

Following changes have been made (Lines 234-244)

Among all monocots and dicots, PHO1 was longer in S. bicolor than in other species. PHO1 had two conserved domains (CDs): GT-35-Glycogen-Phosphorylase (97-950 aa) and a unique L80 domain (500-600 aa). The L80 domain was hyper-variable and predicted to be consists of a PEST motif (a motif of negatively charged aa residues, which could act as a substrate for proteasomes) [31, 32]. Similarly, the N-terminal (1-97 aa) was also hyper-variable, but the linker segments were conserved (Figure 4). Insertions, deletions, and mismatches were the main reasons for variations. Each of the PHO1 is found with the insertion of variable transit peptide of up to 50 amino acids which lack in PHO2 except for the PHO2 of P. halli (Figure S3). The transit peptide contained several structural features common to transit peptides of chloroplast proteins but contains a surprisingly large number of histidine residues. 

Figure 4 have also been updated. An extra figure (Figure S3) has also been included in supplementary materials for sequential features.

  1. Line 248 “....high level of similarity for physico-chemical properties..”. This is confusing in terms of meaning and should be re-written more clearly.

Sorry for this, we have re-written it in the revised version.

Following changes have been made (Lines in revised version 244-246).

In addition, a high similarity level of intrinsic physical and chemical properties was also observed ranging from 56.56% to 97.94% for PHO1 and 70.45% to 99.27% for PHO2 in each of the mentioned species. For PHO1, the root means square deviation (RMSD) value was 2.04–5.68Å in monocots and 2.10–3.87Å in dicots. For PHO2 this was 0.04–1.63Å in monocots and 0.25–0.37Å in dicots.

  1. Figure 7 outlines expression analysis of PHO1 and PHO2 in maize tissues. In Fig 7A more detail is required; please explain what leaf is leaf 1? Is it the first leaf which forms approx. 7days after germination or is it the first leaf harvested from the mature plant when it is flowering? What is the distinction between the ear, seed and endosperm? In Fig 7B the legend states that the data presented is for embryo. Is this correct, or do the authors mean endosperm. If embryo, why was this tissue analyzed and not the starch storing endosperm? If embryo was analyzed, there is no mention in the methods how it was dissected.

Sorry for these several mistakes. We have mentioned the details in the revised version of the manuscript. Leaf1 is actually 1st leaf harvested from a mature plant. The ear is a spike, consisting of a central stem on which tightly packed rows of flowers grow, endosperm tissues was dissected to analyze the transcript level from the 15 DAP seeds. Yes, in figure 7b it was a mistake that was actually endosperm we have revised in updated version.

Following changes have been made (Figure 7 legend)

Figure 7. Distribution and expression pattern of the PHO1 and PHO2 transcripts of Z. mays. (A) Tissue-specific expression levels of the PHO1 and PHO2 of Z. mays. The relative expression pattern has shown and transcript level in embryo is used as control. The leaf1 is the first leaf harvested from the mature plant when it is flowering. The 15 DAP seeds and dissected endosperm were used to analyse the transcript level. (B) Relative expression levels of the PHO1 and PHO2 of Z. mays in different developmental stages of the endosperm. Transcript level in 10 DAP endosperm was used as a control.

  1. Figure 8 shows changes of PHO1 and PHO2 transcript by maize in response to ABA. What tissue in maize was used for these analyses and what developmental stage?

Sorry we forgot to mention this in figure legend. However, in material and method section it was already mentioned. The 15 DAP middle seeds of the corn cob was used to for this analysis.

Following changes have been made.

Figure 8.

Figure 8. Expression level of the PHO1 and PHO2 of maize under ABA treatment. The 15 DAP middle seed of corn cob was used to treat with ABA in MS medium. 0h is indicating no ABA treatment and used as a control. (A) Effect on PHO1 transcripts expression. (B) Effect on PHO2 transcripts expression.

Lines in revised version 524-528

The middle seed of corn cob 15 days after self-pollination of Mo17 was placed in a liquid MS medium containing 50 μmol/L ABA to slowly shake (50 rpm) for 24 h. At the same time, the seeds were cultured in the MS medium without ABA as the control. Some seeds were taken out at indicated time points (0, 1, 4, 12, and 24 h), quickly fro-zen them with liquid nitrogen, and stored for an RT-qPCR experiment.

  1. Line 343 is a typographical error, it should read “hinder”.

Sorry, it was a mistake. The typographical error has removed in the revised version.

Following changes have been made (line in revised version 370)

The 3D modeling suggested the outside arrangements of the L80 domain from the main protein core and was found to restrict the activity of PHO1 for smaller polysaccharide substrates and sterically hinder the large polysaccharides such as starch or glycogen [26].

  1. Please explain the statement on lines 386-388 as to why PHO2 is involved in starch degradation? There is no evidence presented in the current manuscript to support this statement. Lines 388-390 need to be explained. How can PHO1 and PHO2 play the same role in the (starch?) metabolic pathway as they are in different cellular compartments?

Sorry, there were some mistakes in these lines. We have explained and re-write these lines. About the PHO1 and PHO2 same role it was a writing error, we were aiming to write that these enzymes plays identical roles in mentioned species in comparison of PHO1 with PHO1 and PHO2 with PHO2, not same role for each other.

Following changes have been made.

Lines in revised version 415-422

The expression analysis of the PHO2 of Z. mays suggested that it could mainly involve in degradation and maturation of starch as the expression pattern was found relatively higher in lateral stages of endosperm development. These results are agreeable with the PHO2 of H. vulgare [37] and T. aestivum [4] and it was shown that the expression of PHO2 correlates with its activity and it may degrade the reserve starch in plant organs [4]. In addition, it has been suggested previously that it may also be involved in metabolizing the products of starch degradation from chloroplast in cells with intact plastid and regulates cytosolic Glc-1-P levels.

Lines in revised version 422-425

The expression mode of PHO1 was similar in the reported cereal crops, indicating that PHO1 might plays an identical role/activity in the metabolic pathways in each of the examined species as the availability of PHO1 correlates with the expression levels of other starch biosynthetic enzymes in higher plants [35].

  1. There is no explanation as to why both PHO genes are expressed at such low levels in leaves. Some mention of this is warranted, particularly as maize leaves produce starch which is turned overs during a 24h cycle.

Sorry for this mistake, we have included a passage for this explanation in the revised version.

Following addition have been made (Lines in revised version 426-436.)

The expression pattern of both of the isoforms is found relatively low in leaves for each of the examined species. Early speculation that phosphorolysis was the major route of degradation in the chloroplast [49] however, it is now clear that many chloroplasts and starch-containing plastids possess amylases in addition to PHO and can degrade the starch hydrolytically as well as phosphorolytically [50].  The importance of PHO1 and PHO2 in leaves has not been properly characterized. However, a study on the knockout mutant of A. thaliana (lack of PHO1) has found no alteration in starch and sugar metabolism and suggested that hydrolysis rather than phosphorylsis may be the major route of degradation in the chloroplast [51]. In addition, they show that phosphorolysis is not necessary for normal metabolism in most leaf cells.

  1. In the methods section, more detail is required in the section on gene expression analysis. What controls were used. This (control) data should be included.

Sorry, we forgot to mention this. In the revised version we have included more information in the methods section and the figure legends have been mentioned with control also.

Following changes have been made.

Lines in revised version 515-524.

In silico expression analysis of the genes for PHO1 and PHO2 of the main cereal crops (O. sativa, H. vulgare, and A. thaliana) was carried out using the “Genevestigator” microarray database. In the case of Z. mays, the expression analysis in different tissues was carried out using the expression level data determined by RT-qPCR. The extraction of RNA, cDNA synthesis, and expression analysis were carried out as described by Jian M. [37]. The GAPDH gene encoding glyceraldehyde-phosphate dehydrogenase was used as the internal reference. Following primers were used to measure the transcript level of PHO1: Fwd: CTAACAGGACAATATGCA; Rev: GCTTCATTGGCCTTGGCA. Following primers were used to measure the transcript level of PHO2: Fwd: ATGAG-TGCGGCGGCGAA; Rev: CTTGGCAATGCGCTGCTTCA.  

Figure 7.

Figure 7. Distribution and expression pattern of the PHO1 and PHO2 transcripts of Z. mays. (A) Tissue-specific expression levels of the PHO1 and PHO2 of Z. mays. The relative expression pattern has been shown and transcript level in the embryo is used as control. (B) Relative expression levels of the PHO1 and PHO2 of Z. mays in different developmental stages of the endosperm. Transcript level in 10 DAP endosperm is used as a control.

Reviewer 2 Report

The authors need to explain the hypothesis and knowledge gap. It is very important to explain why it is important to study PHO genes in monocot and dicot. Such descriptive work can be done on any plant gene and defiantly they will show exactly the same clustering in the phylogenetic tree. For any plant gene, homologs from monocot will cluster together and dicot will form another cluster. In short, authors need to rewrite the introduction to elaborate on the necessity of their research work.  

In materials and methods - "PHO2 was found except for T. aestivum and H. vulgare. No genomic data about these two species were available in the databases; only a protein sequence was available that was used to generate CDS through the transcription sequence generation online tool." - this is surprising as well as not acceptable. This indicates that the authors do not have sufficient expertise with computational analysis. Both the genomes are available for several years. 

The transcription factor binding sites and the expression profiling does not hold any ground. Nowadays over a thousand transcriptome data is available for each of these species, and authors need to use that to come up with novel findings. 

Author Response

Response Letter to Reviewer

Dear reviewer,

Thank you for your careful reading and positive comments and suggestions upon our manuscript. We have added a point-to-point response to your comments in the revised version of manuscript. We felt that after the incorporation of changes you suggested, manuscript quality has been greatly improved. Moreover, manuscript has been carefully proofread for grammar, punctuation, and English language mistakes.  We have highlighted all the changes we made in red color.

  1. The authors need to explain the hypothesis and knowledge gap. It is very important to explain why it is important to study PHO genes in monocot and dicot. Such descriptive work can be done on any plant gene and defiantly they will show exactly the same clustering in the phylogenetic tree. For any plant gene, homologs from monocot will cluster together and dicot will form another cluster. In short, authors need to rewrite the introduction to elaborate on the necessity of their research work.  

Sorry, some information was missing. As per suggested we have made the changes in the revised version of manuscript.

Following changes have been made in the Introduction.

Maize (Zea mays) is one of the three major cereal crops globally and the largest in food crops with a yield second to rice (Oryza sativa). The major part of the yield is known as starch, which is the most important form of reserve polysaccharide synthesized in cellular organelles in plants called plastids [1]. Starch biosynthesis involves a set of enzymes including ADP-glucose pyrophosphorylase (AGPase; EC 2.7.7.27), starch synthases (SSs; EC 2.4.1.21), branching enzymes (SBEs; EC 2.4.1.18), starch de-branching enzymes [DBEs, 3.2.1.41; mainly isoamylases (ISAs)] and starch phosphorylase (PHO; EC 2.4.1.1) in the cereal endosperm [2, 3]. In recent times, PHO has been investigated comprehensively due to its involvement in starch biosynthesis [4-9].

The PHO is found to be involved in the transfer of glucosyl unit from glucose-1-phosphate (Glc-1-P) to growing α-1-4 linked glucan chain in a reversible reaction that depends on the available substrate level [10]. In plants, two distinct forms of PHO have been reported [11]: plastidial starch phosphorylase (PHO1) and cytosolic starch phosphorylase (PHO2). These two forms have significant sequence similarities to each other but largely differ in their molecular sizes, substrate specificities physiological roles, and intracellular localization [12]. PHO1 resides in the plastid [13] and PHO2 in the cytosol. PHO1 plays an important role only in starch degradation due to the high Pi/Glc-1-P ratio found in vivo [14]. However, no direct evidence is available to support this. Moreover, several starch-degrading enzymes have been identified [15, 16], including glucan water dikinase (GWD; EC 2.7.9.4) and phosphoglucan water dikinase (PWD; EC 2.7.9.5), which produces the branched and linear glucans from the starch [17]. However, α-amylase (EC 3.2.1.1) is thought to be the first enzyme to initiate starch degradation. These findings make the PHO1 an indirect-acting or a regulatory candidate that can influence the activity of other participating enzymes. The exact role of PHO1 is still in debatable. Moreover, PHO1 is reported to be involved in starch metabolism in various monocots and dicots, including Zea mays [18], Oryza sativa [19] Triticum aestivum [4], Hordeum vulgare [20], and Solanum tuberosum [21]. PHO2 can degrade the branched glucans and can even attack the starch granules [22]. In addition, it can use the degraded products from the starch and maintain the Glc-1-P level in the cytosol [23].

These two enzymes are highly similar to each other but differ in the presence of transit peptides in the N-terminal and an extra stretch of 78–82 amino acid (aa) residues (L80 domain) in the middle region of PHO1. The presence of this extra peptide can be responsible for divers biochemical activities [9]. Previously, L80 insertion is found to be an active substrate for proteasomes in Ipomoea batatas [8]. The PHO1 in Solanum tuberosum maintains its intact structure in young tubers but steadily degrades into smaller peptides as the tubers become mature [13]. Both the intact (105 kDa) and de-graded (55 kDa) forms of PHO1 have been reported in Vigna radiata. Interestingly, both forms were identified to make active catalytic complexes [24]. In contrast, the PHO1 of cereals including Oryza sativa [19, 25], Zea mays [18], Hordeum vulgare [26], and Triticum aestivum [10] is not degraded by the proteasome, even though they all have a PEST motif in L80 insertion. A detailed comparative study would be required to uncover these differences to reach a conclusion. No major differences exist between PHO1 and PHO2 in terms of binding Glc-1-P and Pi (inorganic phosphate). However, these can be distinguished by their affinity to bind the glucan substrate (PHO1 has a low and PHO2 has a high affinity for highly branched polysaccharides) [19, 27]. The L80 domain in PHO1 was found to sterically prevent the binding of the enzyme to the branched poly-saccharide substrate [28]. Quantitative data about PHO abundance and activity is lacking and till yet only Hordeum vulgare PHO1 crystal structure has been characterized [26].

Several reports on microarray based analysis combined with promoter sequences indicated that PHO may play a role under different stresses. However, the regulation pattern found variable among main cereal crops, especially in Arabidopsis thaliana where PHO1 suggested to play role in tolerance of abiotic stress rather that degradation of starch [29]. A study on PHO1 of Hordeum vulgare suggested no effect of ABA on the expression of PHO1 but up-regulation of PHO2 [33]. In contrast, in Arabidopsis thaliana significant down-regulation for both isoforms were detected. Considerable in-formation about PHO and its gene is available. However, detailed information about gene diversity, structure variation, function, evolution, and PHO responding to phyto-hormones is unclear in some important crops, including Zea mays. In recent years, significant progress has been made by scientists in sequencing plant genomes. The avail-ability of genome data is essential to answer the biological questions through comparative analysis [30]. Therefore, the present study was carried out using the PHO1 and PHO2 genes of Oryza sativa as reference (as these are well characterized and closely related to Zea mays) to identify and characterize its orthologs from eight important monocots and four important dicots. The obtained results are presented in this communication, including “true” orthologs of PHO1 and PHO2 genes, evolution of gene structure for both isoforms across mentioned species, synteny and collinearity analysis, identification of conserved and variable domains with subtle differences between both isoforms and among the monocots and dicots, promotor analysis, and expression analysis with the effect of ABA treatment on the expression of both isoforms of PHO in Zea mays.

  1. In materials and methods - "PHO2 was found except for T. aestivum and H. vulgare. No genomic data about these two species were available in the databases; only a protein sequence was available that was used to generate CDS through the transcription sequence generation online tool." - this is surprising as well as not acceptable. This indicates that the authors do not have sufficient expertise with computational analysis. Both the genomes are available for several years. 

Sorry, it was a huge mistake we were facing issues to retrieving the data about these orthologs, might because of data upgradation. We had found the data and waiting for the first decision so that we can include this in the revised version. Now the data has been included and each analysis has been updated. The mentioned redundant lines and ideas have also been removed. The changes have also been made in the Tables and figures.

Following changes have been made.

Tables.

Table 1. Details of cDNAs, CDS, genes and protein sequences of PHO1 in different monocots and dicots with respect to O. sativa PHO1.

Species

 PHO1

cDNA

CDS

Gene

*Protein

Length (bp)

% Identical

Length (bp)

% Identical

Length (bp)

% Identical

Length (aa)

% Identical

S. bicolor

3445

79.0

2958

91.9

6536

62.6

985

86.2

Z. mays

3523

78.1

2955

91.9

9022

49.2

984

85.8

S. italica

3385

80.9

2943

92.4

6909

62.8

980

87.2

P. halli

3351

80.7

29.64

91.9

6671

63.0

987

85.8

B. distachyon

3410

81.7

2937

92.5

7498

62.9

978

87.5

T. aestivum

3312

80.0

2917

90.4

7461

58.4

971

84.3

O. sativa

3408

100

2937

100

7239

100

978

100

H. vulgare

3305

77.3

2907

90.6

7441

61.1

968

84.9

A. thaliana

3180

64.5

2889

81.5

5008

45.9

962

69.3

S. lycopersicum

3332

65.0

2901

80.9

8718

46.3

966

67.0

S. tuberosem

4942

42.2

2901

81.6

4942

42.2

966

67.4

C. annuum

3456

63.2

2937

80.5

8233

45.9

978

68.5

* Accession ids for S. bicolor, XP_021306483.1; Z. mays, NP_001296783.1; S. italica, XP_004981704.1; P. halli, XP_025795811.1; B. distachyon, XP_003559211.1; T. aestivum, ACC59201.1; O. sativa, XP_015631420.1; H. vulgare, KAE8783983.1; A. thaliana, Q9LIB2.1; S. lycopersicum, NP_001362574.1; S. tuberosem, NP_001275215.1; C. annuum, XP_016569840.1. - indicates absence of data.

Table 2. Details of cDNAs, CDS, gene, and protein sequences of PHO2 in different monocots and dicots with respect to O. sativa PHO2.

Species

 PHO2

cDNA

CDS

Gene

*Protein

Length (bp)

% Identical

Length (bp)

% Identical

Length (bp)

% Identical

Length (aa)

% Identical

S. bicolor

3167

80.8

2517

92.5

6227

60.6

838

89.5

Z. mays

2868

80.4

2517

92.6

12006

36.7

838

89.9

S. italica

2937

81.7

2514

92.4

6312

61.6

837

89.2

P. halli

2979

81.2

2664

86.0

5902

59.4

897

83.2

B. distachyon

2909

82.5

2502

93.4

6891

62.4

833

90.3

T. aestivum

2856

80.4

2499

93.2

6114

62.2

832

90.3

O. sativa

3036

100

2526

100

6275

100

841

100

H. vulgare

2857

79.4

2499

93.5

5569

59.6

832

90.4

A. thaliana

3168

62.9

2526

85.1

5240

47.6

841

77.6

S. lycopersicum

2908

65.4

2514

84.9

15109

23.7

837

77.1

S. tuberosem

2833

65.9

2517

85.1

13343

29.7

838

77.8

C. annuum

2965

65.0

2544

82.8

34548

11.8

847

75.0

* Accession ids for S. bicolor, XP_002458773.1; Z. mays, NP_001151625.1; S. italica, XP_004970602.1; P. halli, XP_025814328.1; B. distachyon, XP_003564622.1; T. aestivum, Q9LKJ3.1; O. sativa, XP_015621275.1; H. vulgare, KAE8796869.1; A. thaliana, CAB61943.1; S. lycopersicum, XP_004246972.1; S. tuberosem, NP_001275118.1; C. annuum, XP_016542003.1. - indicates absence of data.

Figures 1, 2 and 3 have also been updated.

Lines in revised version 111-116.

The “true” orthologs of the PHO1 and the PHO2-encoding genes of O. sativa were identified from the 12 different species using the criteria mentioned in the Materials and Methods section. Thus, the genome, coding DNA sequence (CDS), and cDNA of PHO1 and PHO2 from the mentioned orthologs were obtained and are presented in Table 1 (PHO1) and Table 2 (PHO2). In all examined species, a single ortholog for each PHO1 and PHO2 was found. The similarity of cDNA of PHO1 in all mentioned orthologs with that of O. sativa (used as reference) ranged from 78.1% to 81.7% in monocots and 42.2% to 65.0% in dicots.

All the supplementary tables have also been updated.

  1. The transcription factor binding sites and the expression profiling does not hold any ground. Nowadays over a thousand transcriptome data is available for each of these species, and authors need to use that to come up with novel findings. 

Thank you for the suggestion, microarray data and transcriptomic data has been searched and significant changes have been made in the revised version of the manuscript and discussed in discussion portion.

Following changes have been made (lines in revised version 404-466).

Several cis-regulatory elements have been prospected previously in the promotor region [32, 37]. We also identified the elements responsible for response to abiotic stress, tissue-specific expression, and light. The light-responsive element might have an important role as the starch synthesis took place during the daytime. Thus, a tight regulation pathway was required to control the response to light. The response elements in the upstream region of PHO1 and PHO2 might be responsible for the high expression level during the endosperm development. The prospected response elements in the present study need to be validated using wet-lab to determine the functional significance.

Microarray-based data have indicated the uniform pattern of PHO1 transcripts in various tissues and seed developmental stages among H. vulgare, O. sativa, and A. thaliana. The higher expression of PHO1 in the developing endosperm during the anthesis and grain filling period and lower expression in vegetative tissues suggested its role in starch metabolism in Z. mays. Similar results were reported in T. aestivum [4], O. sativa [48], and H. vulgare [37]. The expression analysis of the PHO2 of Z. mays suggested that it could mainly involves in degradation and maturation of starch as the expression pattern found relatively higher in lateral stages of endosperm development. These results are agreeable with the PHO2 of H. vulgare [37] and T. aestivum [4] and it was shown that the expression of PHO2 correlates with its activity and it may degrade the reserve starch in plant organs [4]. In addition, it has been suggested previously that it may also be involved in metabolizing the products of starch degradation from chloroplast in cells with intact plastid, and regulates cytosolic Glc-1-P level. The expression mode of PHO1 was similar in the reported cereal crops, indicating that PHO1 might plays the identical role/activity in the metabolic pathways in each of the examined species as the availability of PHO1 correlates with the expression levels of other starch biosynthetic enzymes in higher plants [35]. The expression pattern of both of the isoforms is found relatively low in leaves for each of the examined species. Early speculation that phosphorolysis was the major route of degradation in chloroplast [49] however, it is now clear that many chloroplast and starch containing plastids possesses amylases in addition to PHO and can degrade the starch hydrolytically as well as phosphorolytically [50].  Importance of PHO1 and PHO2 in leaves have not been properly characterized. However, a study on knockout mutant of A. thaliana (lack of PHO1) have found no alteration in starch and sugar metabolism and suggested that hydrolysis rather than phosphorylsis may be the major route of degradation in the chloroplast [51]. In addition, they show that phosphorolysis is not necessary for normal metabolism in most leaf cells.

All studied PHO1 and PHO2 are proposed to have the regions (in promotors) that have functions in ABA-induced expression, but a study on PHO1 of H. vulgare suggested no effect of ABA on the expression of PHO1 but up-regulation of PHO2 [20]. However, in A. thaliana significant down-regulation for both isoforms were detected. In contrast, it is found that ABA can up-regulate the expression of both the PHO1 and PHO2 in Zea mays. The effect of abiotic stresses on the expression of PHO has not been determined for various crops and could be worthy of investigation. For PHO1 and PHO2 in maize, ABA-induced expression changes might be the same according to phylogenetic analysis and aa residue alignments.

Round 2

Reviewer 1 Report

The revised manuscript has satisfied all of my original comments. I have noo further comments to make on this manuscript.

Reviewer 2 Report

Authors have extensively improved the MS and can be published